# Optimistic Gradient Learning with Hessian Corrections for High-Dimensional Black-Box Optimization

## Abstract

Black-box algorithms are designed to optimize functions without relying on their underlying analytical structure or gradient information, making them essential when gradients are inaccessible or difficult to compute. Traditional methods for solving black-box optimization (BBO) problems predominantly rely on non-parametric models and struggle to scale to large input spaces. Conversely, parametric methods that model the function with neural estimators and obtain gradient signals via backpropagation may suffer from significant gradient errors. A recent alternative, Explicit Gradient Learning (EGL), which directly learns the gradient using a first-order Taylor approximation, has demonstrated superior performance over both parametric and non-parametric methods. In this work, we propose two novel gradient learning variants to address the robustness challenges posed by high-dimensional, complex, and highly non-linear problems. Optimistic Gradient Learning (OGL) introduces a bias toward lower regions in the function landscape, while Higher-order Gradient Learning (HGL) incorporates second-order Taylor corrections to improve gradient accuracy. We combine these approaches into the unified OHGL algorithm, achieving state-of-the-art (SOTA) performance on the synthetic COCO suite. Additionally, we demonstrate OHGL's applicability to high-dimensional real-world machine learning (ML) tasks such as adversarial training and code generation. Our results highlight OHGL's ability to generate stronger candidates, offering a valuable tool for ML researchers and practitioners tackling high-dimensional, non-linear optimization challenges.

## 1 Introduction

Black-box optimization (BBO) is the process of searching for optimal solutions within a system's input domain without access to its internal structure or analytical properties Audet et al. (2017c). Unlike gradient-based optimization methods that rely on the calculation of analytical gradients, BBO algorithms query the system solely through input-output pairs, operating agnostically to the underlying function. This feature distinguishes BBO from traditional ML tasks, such as neural network training, where optimization typically involves backpropagation-based gradient computation.

Many real-world physical systems naturally fit into the BBO framework because their analytical behavior is difficult or impossible to model explicitly. In these cases, BBO algorithms have achieved remarkable success in diverse fields, such as ambulance deployment Zhen et al. (2014), robotic motor control Gehring et al. (2014); Prabhu et al. (2018), parameter tuning Olof (2018); Rimon et al. (2024), and signal processing Liu et al. (2020), among others Alarie et al. (2021). BBO applications extend beyond physical systems; many ML problems exhibit a black-box nature when the true gradient is absent. Examples include hyperparameter tuning Bischl et al. (2023), contextual bandit problems Bouneffouf et al. (2020), and large language model training with human feedback Bai et al. (2022), to name a few.

As the scale of data continues to grow, the dimensionality of the problem space increases in tandem. This trend is particularly evident in ML, where model architectures, embedding and latent representation sizes, and the number of hyperparameters are continually expanding. High-dimensional optimization challenges traditional BBO algorithms, which often require the number of collected samples at each step, $n_s$, to scale proportionally with the problem size, $N$. Parametric neural models, however, have shown that reusing past samples can effectively reduce the required sample size such that $n_s \ll N$ Sarafian et al. (2020); Lu et al. (2023). Building on this, we propose a sampling profiler that further reduces the number of samples needed at each step while maintaining performance.

While problem dimensions increase, the cost of evaluating intermediate solutions remains a critical constraint, especially in real-world settings where interaction with the environment is expensive or in ML tasks where larger models counterbalance gains in computational power. Therefore, modern BBO algorithms must not only reduce evaluation steps but also converge more quickly Hansen et al. (2010). Achieving this requires algorithms capable of more accurately predicting optimization directions, either through better gradient approximation Anil et al. (2020); Lesage-Landry et al. (2020) or momentum-based strategies to handle non-convexity and noise. In this paper, we propose two key improvements to Explicit Gradient Learning (EGL): (1) Optimistic Gradient Learning (OGL), a weighted gradient estimator that biases toward promising solutions and (2) Higher-Order Gradient Learning (HGL), which incorporates Hessian corrections to yield more accurate gradient approximations.

We combine the strengths of OGL and HGL to a unified algorithm termed OHGL which exhibits four key advantages:

- **Robustness**: OHGL consistently outperforms baseline algorithms across a diverse range of benchmark problems, including synthetic test suites and real-world ML applications. Its ability to handle noisy and non-convex environments.

- **Gradient Precision**: By integrating the second-order information via Hessian corrections, OHGL achieves significantly more accurate gradient approximations than standard EGL.

- **Convergence Rate**: OHGL demonstrates faster convergence rate.

- Utilizing the sampling profiler and the optimistic approach, OGL is able to solve **high-dimensional problems** with smaller budget and **converge faster** than baseline algorithms.

**Related works:** Black-box optimization (BBO) algorithms have a long history, with various approaches developed over the years. Some of the foundational techniques include grid search, coordinate search Audet et al. (2017e), simulated annealing Busetti (2003), and direct search methods like Generalized Pattern Search and Mesh Adaptive direct search Audet et al. (2017a), Gradient-less descent Golovin et al. (2019), and ZOO Chen et al. (2017). These approaches iteratively evaluate potential solutions and decide whether to continue in the same direction. However, they resample for every step and don't use the sampled budget from previous iterations, wasting a lot of budget.

Another prominent family of BBO algorithms is the genetic algorithm family Back (1996). This includes methods such as Covariance Matrix Adaptation (CMA) Hansen (2016) and Particle Swarm Optimization (PSO) Clerc (2010). These algorithms simulate the process of natural evolution, where a population of solutions evolves through mutation and selection Audet et al. (2017b). They are considered state-of-the-art (SOTA) in optimization due to their effectiveness in tackling complex problems. However, they come with significant drawbacks, particularly the need for extensive fine-tuning of parameters like generation size and mutation rates. CMA, for example, struggles in higher-dimensional environments and requires careful adjustment of hyperparameters and guidance to perform optimally Loshchilov et al. (2013); Tang (2021). In this work, we propose a simpler method to enhance the performance of CMA, particularly in high-dimensional settings.

Then there are model-based methods Audet et al. (2017d), which attempt to emulate the behavior of the function using a surrogate model. These models provide important analytical information, such as gradients Bertsekas (2015), to guide the optimization process and help

find a minimum. Within this class, we can further distinguish two sub-classes. To address the issue of dimensionality, Explicit Gradient Learning (EGL) was proposed by Sarafian et al. (2020). While many model-based methods focus on learning the function's structure to derive analytical insights (e.g., Indirect Gradient Learning or IGL Lillicrap (2015); Sarafian et al. (2020)), EGL directly learns the gradient information. EGL uses Taylor's theorem to estimate the gradient. The authors also emphasize the importance of utilizing a trust region to handle black-box optimization problems. However, EGL has some drawbacks: it often uses the available budget inefficiently, disregarding both the complexity and dimensionality of the environment. Additionally, the datasets created by EGL can be naive, leading to over-fitting or improper network learning. This work tackles these issues by showing the importance of proper algorithm calibration and optimization.

Recent work also highlights the limitations of common assumptions in BBO algorithms, such as continuity or Gaussian distributions of functions, which can hinder optimization. For instance, OPT-GAN Tang (2021), a generative model, seeks to bypass these assumptions by learning a function's distribution and generating better candidate solutions based on that knowledge.

**The paper is organized as follows:** Section 2 covers the algorithm's theoretical background and mathematical foundations. Sections 3 and 4 present our two enhanced variants of the gradient learning algorithm: OGL and HGL, these are followed by section 5 where we present the full algorithm OHGL. Section 6 provides experimental results on the synthetic COCO test suite and Section 7 highlights 2 real-world high-dimensional applications and potential uses. Finally, section 8 concludes and suggests future research directions. Our code, experiments, and environment setup are available in the supplementary material.

## 2 BACKGROUND

The goal of black-box optimization (BBO) is to minimize a target function $f(x)$ through a series of evaluations Audet et al. (2017c), over a predefined domain $\Omega$:

$$\text{find: } x^* = \arg\min_{x \in \Omega} f(x) \tag{1}$$

The Explicit Gradient Learning method, as proposed by Sarafian et al. (2020) leverages the first-order Taylor's expansion: $f(y) = f(x) + \nabla f(x)^\top (y - x) + R_1(x, y)$. Here, $R_1(x, y) = O(\|y - x\|^2)$ is a higher-order residual. By minimizing the residual term with a surrogate neural network model, EGL learns the *mean-gradient*: a smooth approximation of the function's gradient

$$g_\varepsilon^{EGL}(x) = \arg\min_{g_\theta : \mathbb{R}^n \to \mathbb{R}^n} \int_{y \in B_\varepsilon(x)} \left( f(x) - f(y) + g_\theta(x)^\top (y - x) \right)^2 dy \tag{2}$$

where $B_\varepsilon(x)$ is a ball around $x$. As $\varepsilon \to 0$, the mean-gradient converges to $\nabla f$, this property lets EGL explore for lower regions in the function landscape when $\varepsilon$ is sufficiently large and converge to a local minimum when $\varepsilon$ converges to 0.

A key component of the EGL algorithm is the *trust region* (TR), which restricts the search space around the current estimate. This region standardizes input-output statistics, enhancing the neural network's effectiveness. Although Sarafian et al. (2020) suggested the TR framework as part of the EGL algorithm, TR is not exclusive to EGL and can significantly improve other algorithms. In our work, we created stronger baseline algorithms by adding TR

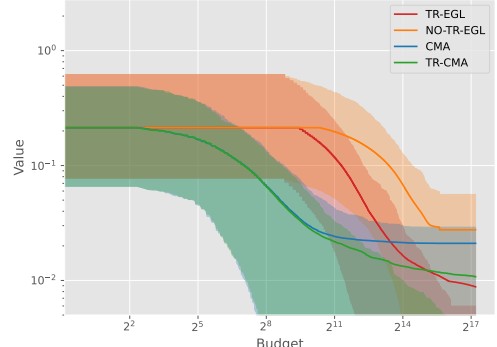

Figure 1: Comparing the effect of trust region. Trust region significantly improves both EGL and CMA algorithms.

to the classic Covariance Matrix Adaptation (CMA) algorithm (see Appendix: Algorithm

11). Notably, this seemingly minor change to CMA significantly improves its performance and outperforms vanilla EGL in the COCO optimization suite, as shown in Fig. 1.

## 3 Optimistic Gradient Learning with Weighted Gradients

While local search algorithms like EGL are based on the notion that the gradient descent path is the optimal search path, following the true gradient path may not be the optimal approach in terms of sampling budget utilization, avoiding shallow local minima and navigating through narrow ravines. Specifically, in case we obtain a batch $D$ of sample pairs $\{(x_i, f(x_i))\}_{i \in D}$ around our current solution $x$, a plausible and optimistic heuristic can be to direct the search path towards low regions in the sampled function landscape regardless the local curvature around $x$. In other words, to take a sensible guess and bias the search path towards the lower $(x_i, f(x_i))$ samples. To that end, we define the Optimistic Gradient Learning objective by adding an *importance sampling* weight to the integral of Eq. 2

$$g_\varepsilon^{OGL}(x) = \underset{g_\theta:\mathbb{R}^n \to \mathbb{R}^n}{\arg\min} \int_{y \in B_\varepsilon(x)} W_f(x,y) \cdot \left(f(x) - f(y) + g_\theta(x)^\top (y-x)\right)^2 dy \qquad (3)$$

Here, $W_f$ is a softmax-like weight function that normalizes the weight by the sum of exponents across the sampled batch

$$W_f(x,y) = \frac{e^{-\min(f(x),f(y))}}{\sum_{x_i \in D} e^{-f(x_i)}} \qquad (4)$$

Notice that in practice, the theoretical objective in Eq. 3 is replaced by a sampled Monte-Carlo version (see Sec. 5) s.t. the sum of all weights across the sampled batch is smaller than 1. In the following theorem, we show that the *controllable accuracy* property of EGL, which implies that the mean-gradient converges to the true gradient still holds for our biased version, s.t. when $\varepsilon \to 0$, $g_\varepsilon^{OGL} \to \nabla f(x)$ this guarantees that the convergence properties of the mean-gradient still hold

**Theorem 1** *(Optimistic Controllable Accuracy) For any differentiable function $f$ with a continuous gradient, there exists $\kappa^{OGL} > 0$ such that for any $\varepsilon > 0$, $g_\varepsilon^{OGL}(x)$ satisfies*

$$\|g_\varepsilon^{OGL}(x) - \nabla f(x)\| \leq \kappa^{OGL} \varepsilon \quad \text{for all } x \in \Omega.$$

In Fig. 2a we plot 4 typical trajectories of EGL and OGL which demonstrate why in practice and on average we can benefit from biasing the gradient. We find that the biased version avoids getting trapped in local minima and avoids long travels through narrow ravines which rapidly consume the sampling budget. In Fig. 2b we find statistically that OGL tends to progress faster and closer to the global minimum while EGL diverges from the global minimum in favor of local minima.

## 4 Gradient Learning with Hessian Corrections

To learn the mean-gradient, EGL minimizes the first-order Taylor residual (see Sec. 2). Utilizing higher-order approximations has the potential of learning more accurate models for the mean-gradient. Specifically, the second-order Taylor expansion is

$$f(y) = f(x) + \nabla f(x)^\top (y-x) + \frac{1}{2}(y-x)^\top \nabla^2 f(x)(y-x) + R_2(x,y) \qquad (5)$$

Here $R_2(x,y) = O(\|x-y\|^3)$ is the second order residual. Like in EGL, we replace $\nabla f$ with a surrogate model $g_\theta$ and minimize the surrogate residual to obtain our Higher-order Gradient Learning (HGL) variant

$$g_\varepsilon^{HGL}(x) = \underset{g_\theta:\mathbb{R}^n \to \mathbb{R}^n}{\arg\min} \int_{y \in B_\varepsilon(x)} \mathcal{R}_{HGL}^2(x,y) dy$$

$$\mathcal{R}_{g_\theta}^{HGL}(x,y) = f(x) - f(y) + g_\theta(x)^\top (y-x) + \frac{1}{2}(y-x)^\top J_{g_\theta}(x)(y-x) \qquad (6)$$

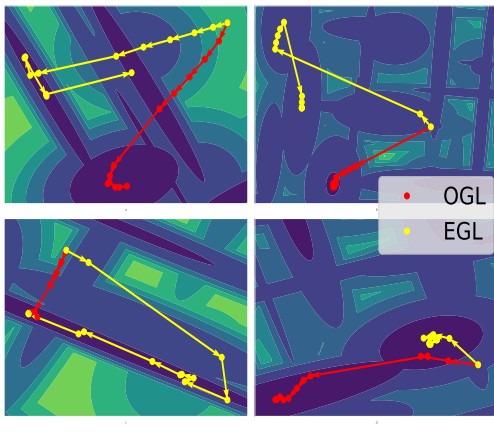 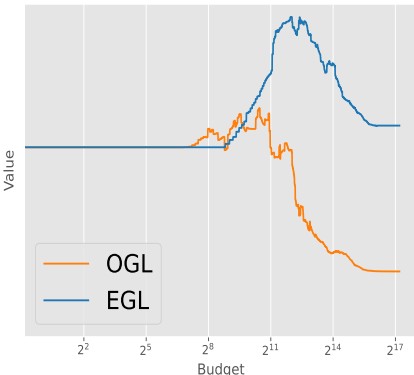

(a) OGL vs EGL trajectories. 1st row: Gallagher's Gaussian 101-me. 2nd row: 21-hi.

(b) The average Euclidean distance at each step between the algorithm and the global minimum.

Figure 2: Training with the mean-gradient EGL versus the optimistic version OGL.

The new higher-order term $J_{g_\theta}(x)$ is the Jacobean of $g_\theta(x)$, evaluated at $x$ which approximates the function's Hessian matrix in the vicinity of our current solution, i.e., $J_{g_\theta}(x) \approx \nabla^2 f(x)$. Next, we show theoretically that as expected, HGL converges faster to the true gradient which amounts to lower gradient error in practice.[1]

**Theorem 2** *(Improved Controllable Accuracy): For any twice differentiable function $f \in \mathcal{C}^2$, there exists $\kappa_{HGL} > 0$ such that for any $\varepsilon > 0$, the second-order mean-gradient $g_\varepsilon^{HGL}(x)$ satisfies*

$$\|g_\varepsilon^{HGL}(x) - \nabla f(x)\| \le \kappa_{HGL}\varepsilon^2 \quad \text{for all } x \in \Omega.$$

In other words, in HGL the model error is in an order of magnitude of $\varepsilon^2$ instead of $\varepsilon$ in EGL and OGL. In practice, we verified that this property of HGL translates to more accurate gradient models. Figure 3 shows the gradient error of EGL and HGL for a selected set of problems from the COCO test suite where the analytical true gradient can be easily calculated and compared to our learned model.

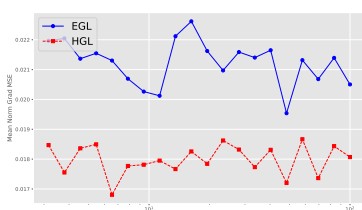

Learning the gradient with the Jacobian corrections introduces a computational challenge as double back-propagation can be expensive. This overhead can hinder the scalability and practical application of the method. A swift remedy is to detach the Jacobian matrix from the competition graph. While this step slightly changes the objective's gradient (i.e. the gradient trough ($R_{HGL}$) it removes the second-order derivative and in practice, we found that it achieves almost similar results compared to the full backpropagation through the residual $R_{HGL}$, see Fig. 4(b).

Figure 3: Comparing the normalized MSE between the true gradient and EGL and HGL gradient models

# 5 OHGL

In this section, we combine the optimistic approach from Sec. 3 and the Hessian corrections from Sec. 4. However, unlike previous sections, we will present the practical implementation

---

[1]Notice that, while HGL incorporates Hessian corrections during the gradient learning phase, we deliberately avoid modifying the gradient descent step to include inverse Hessian scaling, as is done in Newton's methods. In practice, inverting the approximated Hessian matrix can lead to numerical challenges and instabilities and was found to be less effective in our experiments.

where integrals are replaced with Monte-Carlo sums of sampled pairs. In this case, the loss function which is used to obtain the Optimistic Higher-order Gradient (OHGL) model is

$$\mathcal{L}_\varepsilon^{OHGL}(\theta) = \sum_{\|x_i - x_j\| \leq \varepsilon} W_f(x_i, x_j) \times$$

$$\left( f(x_i) - f(x_j) + g_\theta(x_i)^\top (x_j - x_i) + \frac{1}{2}(x_j - x_i)^\top J_{g_\theta}(x_i)(x_j - x_i) \right)^2 \quad (7)$$

The summation is applied over sampled pairs which satisfy $\|x_i - x_j\| \leq \varepsilon$. As explained in Sec. 4, we detach the Jacobian of $g_\theta$ from the computational graph to avoid second-order derivatives. Our final algorithm is outlined in Alg. 1 and an extended version including additional technical details is found in the appendix (See Alg. 2).

---

**Algorithm 1** OHGL (Optimistic Higher-Order Gradient Learning)

---

**Require:** $x_0, \Omega, \alpha, \epsilon_0, \gamma_\alpha, \gamma_\epsilon, n_{\max}, \lambda, \text{Budget}$

$k = 0, j = 0, \Omega_j \leftarrow \Omega$

**while** Budget $> 0$ **do**

    **Explore:** Generate and evaluate samples $\mathcal{D}_k = \{\tilde{x}_i, y_i\}_{i=1}^m$

    **Create Dataset:** Select $m$ tuples $\mathcal{T}$ from $\mathcal{D}_k$

    **Weighted Output:** Assign weights $w_i$ and apply squashing function $\tilde{y}_i = r_k(w_i y_i)$

    **Higher-Order Gradient Learning:** Minimize the loss of $\theta_k$ (Eq. 7)

    **Gradient Descent:** Update solution: $x_{k+1} \leftarrow x_k - \alpha \tilde{g}_{\theta_k}(x_k)$

    **if** $f(h_j^{-1}(\tilde{x}_{k+1})) > f(h_j^{-1}(\tilde{x}_k))$ for $n_{\max}$ times **then**

        Generate new trust region $\Omega_{j+1}$ and update $\epsilon_j$

    **end if**

    **if** $f(h_j^{-1}(\tilde{x}_k)) < f(h_j^{-1}(\tilde{x}_{\text{best}}))$ **then**

        Update $x_{\text{best}}$

    **end if**

    $k \leftarrow k + 1; \text{Budget} = \text{Budget} - m$

**end while**

**return** $x_{\text{best}}$

---

## 6 EXPERIMENTS IN THE COCO TEST SUITE

We evaluated the OGL, HGL and OHGL algorithms on the COCO framework Hansen et al. (2021) and compared them to EGL and other strong baselines: (1) CMA and its trust region variants L-CMA (linear trust region mapping function) and T-CMA (tanh trust region mapping function) and (2) Implicit Gradient Learning (IGL) Sarafian et al. (2020) where we follow the EGL protocol but train a model for the objective function and obtain the gradient estimation by backpropagation as in DDPG Lillicrap (2015). We also adjusted EGL hyper-parameters A.4 and improved the trust region A.5 to reduce the budget usage by our algorithms.

We use the following evaluation metrics:

- **Convergence Rate**: Speed of reaching the global optimum.
- **Success Rate**: Percentage of problems solved within a fixed budget.
- **Robustness**: Performance stability across different hyperparameter settings.

Performance was normalized against the best-known solutions to minimize bias: `normalized_value` $= \frac{y - y_{min}}{y_{max} - y_{min}}$. A function was considered solved if the normalized value was below 0.01.

### 6.1 SUCCESS AND CONVERGENCE RATE

Figure 4(c) illustrates the success rate of each algorithm relative to the distance from the best point required for solving a function. The results show that both OGL and OHGL

| Metric | T-CMA | L-CMA | EGL | **OGL** | **HGL** | **OHGL** |
|---|---|---|---|---|---|---|
| Budget to reach 0.01 | $17,828 \pm 32648$ | $\mathbf{6497 \pm 19731}$ | 24,102 | $8981 \pm 9177$ | $22,146 \pm 29757$ | $26,706 \pm 32676$ |
| Mean | $0.01 \pm 0.05$ | $0.01 \pm 0.05$ | $0.01 \pm 0.03$ | $0.003 \pm 0.02$ | $0.006 \pm 0.03$ | $\mathbf{0.002 \pm 0.02}$ |
| Solved Functions | $0.86 \pm 0.023$ | $0.88 \pm 0.023$ | $0.83 \pm 0.023$ | $0.92 \pm 0.022$ | $0.89 \pm 0.022$ | $\mathbf{0.926 \pm 0.022}$ |

Table 1: Comparison of different metrics: Budget used to reach 0.99 of the final score ($\downarrow$), the mean normalized results ($\downarrow$), std ($\downarrow$); and percentage of solved problems ($\uparrow$).

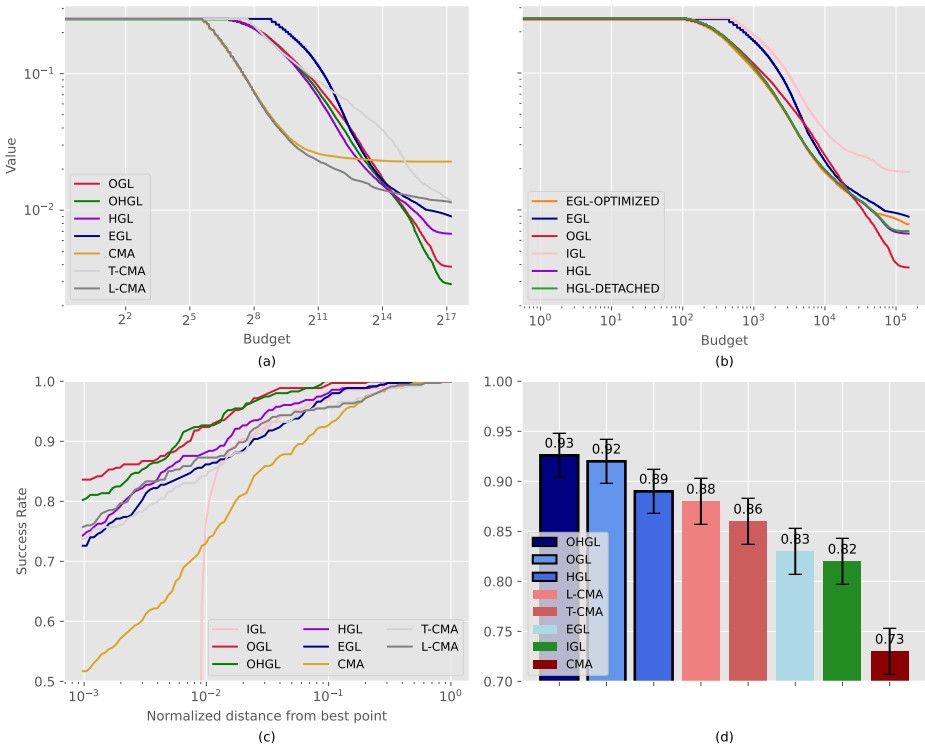

Figure 4: Experiment results against the baseline: (a) Convergence for all our algorithms against baseline algorithms, (b) ablation test for EGL enhancements, (c) Success rate of algorithms as a function of the normalized distance from the best-known solution, (d) Percentage of solved algorithms when the distance from the best point is 0.01

consistently outperform all other algorithms. In particular where the error should be small ($<0.01$), with t-tests yielding p-values approaching 1, indicating strong statistical confidence (see the complete list of t-test p-values in Table 4 in the appendix).

Figure 4(a) presents the convergence rates for the seven algorithms, where OGL, HGL, and OHGL demonstrate superior convergence compared to the other methods. OHGL, in particular, consistently ranks among the top performers across most metrics, as seen in Table 1, which compares multiple performance indicators. Although OGL achieves better initial results, as it searches longer for the space with the optimal solution, making it a worse fit for problems with a low budget, it ultimately reaches superior results. Additionally, Figure 4(b) shows an ablation test, confirming that the core components of our work: the optimistic approach (OGL) and the Higher-order corrections (HGL) contributed substantially much more to the overall performance than other technical improvements to the algorithm (with respect to vanilla EGL), these technical improvements are denoted as EGL-OPTIMIZED and they amount to better trust-region management and more efficient sampling strategies which mainly helps in faster learning rate at the beginning of the process (as described in Sec. A.4 and A.5 in the appendix).

## 6.2 Hyperparameter Tolerance

In this part, we evaluate the robustness of our algorithm to hyperparameter tuning. Our objective is to demonstrate that variations in key hyperparameters have minimal impact on the algorithm's overall performance. To assess this, we conducted systematic experiments, modifying several hyperparameters and analyzing their effects. Table 2 (and Table 6 in the appendix) reports the coefficient of variation (CV), defined as $CV = \frac{\sigma}{\mu}$, across different hyperparameter sweeps, highlighting the algorithm's stability under varying conditions.

Our findings indicate that certain hyperparameters—such as the epsilon factor, shrink factor, and value normalizer learning rate (LR)—exhibit cumulative effects during training. While small variations in these parameters may not have an immediate impact, their influence can accumulate over time, potentially leading to significant performance changes. In A.11.2, we establish the relationship between the step size and the epsilon factor necessary for ensuring progress toward a better optimal solution. When selecting these parameters, this relationship must be considered. Additionally, the shrink factor for the trust region should be chosen relative to the budget, enabling the algorithm to explore the maximum number of sub-problems. This underscores the importance of fine-tuning these parameters for optimal results. Conversely, we found that the structural configuration of the neural network, including the number and size of layers, had minimal effect on performance. This suggests that the algorithm's reliance on Taylor loss enables effective learning even with relatively simple network architectures, implying that increasing model complexity does not necessarily yield substantial improvements.

| Metric | Networks | Epsilon | Epsilon Factor | Training Bias | Perturbation |
|---|---|---|---|---|---|
| CV | 0.0167 | 0.0339 | 0.2361 | 0.1292 | 0.1642 |
| Metric | TR Shrink Method | Normalizer LR | Normalizer Outlier | TR Shrink Factor | Optimizer LR |
| CV | 0.0333 | 0.1618 | 0.0333 | 0.4894 | 0.5625 |

Table 2: Coefficient of Variation ($CV = \frac{\sigma}{\mu}$) over a Hyperparameter sweep experiment.

# 7 High Dimensional Applications

## 7.1 Adversarial Attacks

As powerful vision models like ResNet Targ et al. (2016) and Vision Transformers (ViT) Han et al. (2022) grow in prominence, adversarial attacks have become a significant concern. These attacks subtly modify inputs, causing models to misclassify them, while the perturbation remains imperceptible to both human vision and other classifiers Tang et al. (2019). Formally, an adversarial attack is defined as:

$$x_a^* = \arg\min_x d(x, x_a) \quad \text{s.t.} f(x) \neq f(x_a) \tag{8}$$

Where $f$ is the classifier and $d$ is some distance metric between elements.

Recent studies have extended adversarial attacks to domains like AI-text detection Sadasivan et al. (2024) and automotive sensors Mahima et al. (2024). These attacks prevent tracking and detection, posing risks to both users and pedestrians. Adversarial attacks are classified into black-box and white-box methods. Black-box attacks only require query access, while white-box methods use model gradients to craft perturbations Machado et al. (2021); Cao et al. (2019). Despite some black-box methods relying on surrogate models Dong et al. (2018); Xiao et al. (2018); Madry et al. (2017); Goodfellow et al. (2014), approaches like AutoZOOM Tu et al. (2019) generate random samples to approximate gradient estimation, though they are computationally expensive. Other methods use GAN networks to search latent spaces for adversarial examples Liu et al. (2021); Sarkar et al. (2017). Still, they depend on existing GANs and their latent space diversity.

Our Enhanced Gradient Learning method offers a true black-box approach with precise perturbation control, avoiding gradient back-propagation. OGL directly optimizes perturbations to maintain low distortion while fooling the model, handling high-dimensional spaces with over 30,000 parameters.

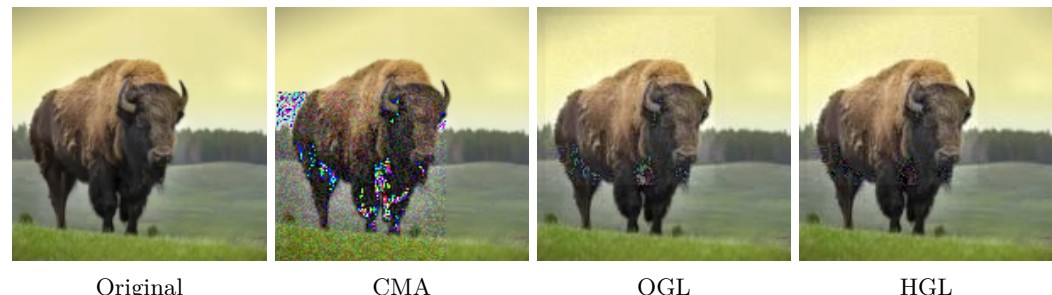

| Original | CMA | OGL | HGL |

Figure 5: Adversarial examples generated by OGL and CMA against the ImageNet model.

**Methodology**: We applied OGL to classifiers trained on MNIST, CIFAR-10, and ImageNet, aiming to minimize equation 8. To generate adversarial images with minimal distortion, we developed a penalty that jointly minimizes MSE and CE-loss A.9. This approach successfully fooled the model, evading the top 5 classifications.

**Results**: We evaluated four different configurations: CMA, OGL, HGL, and a combination of both (CMA+OGL) where the CMA run provides the initial guess for an OGL run. While CMA alone was not able to converge to a satisfying adversarial example, the combined CMA+OGL enjoyed the rapid start of CMA with the robustness of OGL s.t. it was able to find a satisfying adversarial example in 50% of the time used by OGL and HGL.

## 7.2 CODE GENERATION

The development of large language models (LLMs) such as Transformers Vaswani (2017) have advanced code generation Dehaerne et al. (2022). Despite these strides, fine-tuning outputs based on parameters measured post-generation remains challenging. Recent algorithms have been developed to generate code tailored for specific tasks using LLMs. For instance, FunSearch Romera-Paredes et al. (2024) generates new code solutions for complex tasks, while Chain of Code Li et al. (2023) incorporates reasoning to detect and correct errors in the output code. Similarly, our method uses black-box optimization to guide code generation for runtime efficiency. Building on Zhang et al. (2024), which links LLM expertise to a small parameter set, we fine-tuned the embedding layer to reduce Python code runtime. Using LoRA Hu et al. (2021), we optimized the generated code based on execution time, scaling up to $\sim$ 200k parameters.

**Fibonacci:** We tested this approach by having the model generate a Fibonacci function. Initially incorrect, optimization guided the model to a correct and efficient solution. Figure 1 illustrates this progression, with the 25th step showing an optimized version.

**Line-Level Efficiency Enhancements:** We tested the model's ability to implement small code efficiencies, such as replacing traditional for-loops with list comprehensions. The algorithm optimized the order of four functions—'initialize', 'start', 'activate', and 'stop'—each with eight variants, minimizing overall runtime by optimizing the function order.

**Code Force Challenge:** For a more complex problem, we used the Count Triplets challenge from Codeforces[2]. While the model initially struggled, once it found a correct solution, the algorithm further optimized it for runtime performance A.3.

---

[2]https://codeforces.com/

| Metric | CMA | OGL | HGL | CMA+OGL |
|---|---|---|---|---|
| Accuracy | 0.02 | 0.02 | 0.02 | 0.02 |
| MSE | 0.05 | **0.001** | 0.002 | 0.001 |
| Time (Until Convergence) | **20m** | 6H | 7H | 3H |

Table 3: Comparison of methods on Accuracy, MSE, and Time.

```
def fib(n):
    """
    Returns the n number in
    ↪   the Fibonacci series
    """
    return fib1(n-1) + fib2(n)
```

**Step 1 - Starting point**

```
def fib(n):
    """
    Returns the n number in
    ↪   the Fibonacci series
    """
    if n==1:
        return 0
    if n==2:
        return 1
    return fib(n-1) + fib(n-2)
```

**Step 7 - First correct code**

```
def fib(n):
    """
    Returns the n number in
    ↪   the Fibonacci series
    """
    if n <= 1:
        return n
    a, b = 0, 1
    for i in range(2, n + 1):
        a, b = b, a + b
    return b
```

**Step 25 - Convergence**

Figure 6: Generated code samples by OGL algorithm

**Discussion:** Our method demonstrates the ability to generate correct solutions while applying micro-optimizations for efficiency. In simple tasks like Fibonacci, OGL converged on an optimal solution 7, and in more complex problems, it improved the initial solutions. However, in more complex tasks, the LLM may generate code that fails to solve the problem, hindering the optimization of the run-time. To address this, either stronger models or methods focused on optimizing solution correctness are needed. This would ensure that valid solutions are generated first, which can then be further optimized for performance.

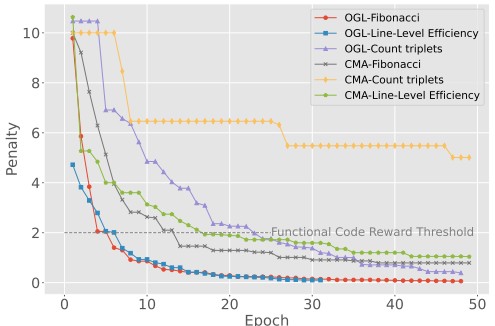

Figure 7: Code generation experiments: penalty over time.

## 8 CONCLUSION

In this work, we introduced several key enhancements to the EGL algorithm, resulting in OHGL, a powerful black-box optimization tool capable of addressing various complex problems. We demonstrated improvements such as second-order gradient approximation, optimistic gradients, and trust region enhancements, which collectively led to faster convergence and more robust performance, particularly in high-dimensional and intricate tasks.

Our experiments showcased OHGL's superiority over state-of-the-art methods, especially in tasks where computational efficiency and precision are paramount, such as adversarial attacks on vision models and optimizing code generation. OHGL's ability to navigate complex optimization spaces while minimizing computational cost demonstrates its utility across various applications.

Looking ahead, Future research should explore new applications of black-box optimization algorithms like OHGL. While OHGL has shown promise in high-dimensional tasks, its efficiency could be further improved. Our research utilized dimension-reduction methods such as Hu et al. (2021). Similar techniques (Zhao et al. (2024); Anil et al. (2020)) can help calculate the Hessian more efficiently.

In conclusion, OHGL represents a significant advancement in BBO algorithm design, providing a robust framework for solving complex optimization problems. However, the true potential of BBO algorithms lies in their broader applicability, and future research should focus on unlocking new avenues for their use, especially in generative models and other cutting-edge areas.

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

## A   APPENDIX

### A.1   FULL OHGL ALGORITHM

---

**Algorithm 2** OHGL (Hessian Weighted Gradient Learning)

---

**Require:** $x_0, \Omega, \alpha, \epsilon_0, \gamma_\alpha < 1, \gamma_\epsilon < 1, n_{\max}, \lambda$

  $k = 0$
  $j = 0$
  $\Omega_j \leftarrow \Omega$
  Map $h_0 : \Omega \rightarrow \mathbb{R}^n$
  Map $W_0 : \mathbb{R} \rightarrow \mathbb{R}$
  **while** Budget $> 0$ **do**
    **Explore:**
      Generate samples $\mathcal{D}_k = \{\tilde{x}_i\}_{i=1}^m, \tilde{x}_i \in V_{\epsilon_k}(\tilde{x}_k)$
      Evaluate samples $y_i = f(h_0^{-1}(\tilde{x}_i)), \ i = 1, \dots, m$
      Add tuples to the replay buffer: $\mathcal{D} = \mathcal{D} \cup \mathcal{D}_k$
    **Create Dataset of Tuples:**
      $\mathcal{T} \leftarrow \{(\tilde{x}_i, \tilde{x}_j) \mid \|\tilde{x}_i - \tilde{x}_j\|_2 < \epsilon, \forall i, j\}$
      Select the first $m$ tuples from $\mathcal{T}$
    **Weighted Output Map:**
      Assign weights $w_i = W_k(y_i)$ to samples based on function values
      Apply squashing function to the outputs: $\tilde{y}_i = r_k(w_i y_i), \ i = 1, \dots, m$
    **Higher-Order Gradient and Hessian Learning:**
      Calculate the Hessian from the Jacobian of the network: $H_k(x) = \mathrm{J}(g_{\theta_k}(x))$
      Optimize the network with GD using the formula in 7
    **Gradient Descent:**
      Update the current solution using second-order information:
      $x_{k+1} \leftarrow x_k - \alpha \cdot \tilde{g}_{\theta_k}(x_k)$
    **if** $f(h_j^{-1}(\tilde{x}_{k+1})) > f(h_j^{-1}(\tilde{x}_k))$ for $n_{\max}$ times in a row **then**
      Generate a new trust region: $\Omega_{j+1} = \gamma_\alpha |\Omega_j|$ with center at $x_{\text{best}}$
      Map $h_j : \Omega \rightarrow \mathbb{R}^n$
      Map $W_j : \mathbb{R} \rightarrow \mathbb{R}$
      $j \leftarrow j + 1$
      $\epsilon_j \leftarrow \gamma_\epsilon \epsilon_j$
    **end if**
    **if** $f(h_j^{-1}(\tilde{x}_k)) < f(h_j^{-1}(\tilde{x}_{\text{best}}))$ **then**
      $x_{\text{best}} = h_j^{-1}(\tilde{x}_k)$
    **end if**
    $k \leftarrow k + 1; \text{Budget} \leftarrow \text{Budget} - m$
  **end while**
  **return** $x_{\text{best}}$

---

## A.2 STATISTICAL ANALYSIS

Table 4: T-test of our algorithms against the benchmark in dimension 40

| Metric | OGL | HOGL | HGL |
|--------|----------|----------|----------|
| IGL | 1.000000 | 1.000000 | 1.000000 |
| OGL | 0.500000 | 0.278378 | 0.334161 |
| OHGL | 0.721622 | 0.500000 | 0.473754 |
| HGL | 0.665839 | 0.526246 | 0.500000 |
| EGL | 0.999961 | 0.999975 | 0.999098 |
| CMA | 1.000000 | 1.000000 | 1.000000 |
| T-CMA | 0.999934 | 0.999911 | 0.999788 |
| L-CMA | 0.999859 | 0.999843 | 0.998842 |

## A.3 GENERATED CODE FROM CODE FORCE EXAMPLE

```python
def count_triplets(n: int, x: int) -> int:
    """
    This function calculates the number of triplets (a, b, c) of positive integers such that:
    1. a*b + a*c + b*c â¤ n
    2. a + b + c â¤ x
    where the order of triplets matters (e.g., (1, 1, 2) and (1, 2, 1) are different triplets)
    and a, b, c must be strictly greater than 0.
    """
    pass
def check_function():
    assert count_triplets(10, 5) == 3, 'Test case 1 failed'
    assert count_triplets(15, 10) == 6, 'Test case 2 failed'
    assert count_triplets(20, 15) == 10, 'Test case 3 failed'
    assert count_triplets(100, 50) == 50, 'Test case 4 failed'
    print('All test cases passed!')
```

Figure 8: Step 1

Figure 9: Generated code samples by the algorithm

```python
def count_triplets(n):
    """
    This function calculates the number of triplets (a, b, c) of positive integers such that:
    1. a*b + a*c + b*c <= n
    2. a + b + c <= x
    where the order of triplets matters (e.g., (1, 1, 2) and (1, 2, 1) are different triplets)
    and a, b, c must be strictly greater than 0.
    """
    count = 0
    # Iterate through all possible values of a, b, c
    for a in range(1, x + 1):
        for b in range(1, x + 1):
            for c in range(1, x + 1):
                if a * b + a * c + b * c <= n and a + b + c <= x:
                    count += 1
    return count
```

Figure 10: Step 21

Figure 11: Generated code samples by the algorithm

```python
def count_triplets(n):
    """
    This function calculates the number of triplets (a, b, c) of positive integers such that:
    1. a*b + a*c + b*c <= n
    2. a + b + c <= x
    where the order of triplets matters (e.g., (1, 1, 2) and (1, 2, 1) are different triplets)
    and a, b, c must be strictly greater than 0.
    """
    count = 0
    # Iterate over a
    for a in range(1, x + 1):
        # Iterate over b
        for b in range(1, x - a + 1):
            # Find the maximum c based on the condition a + b + c <= x
            max_c = min(x - a - b, (n - a * b) // (a + b))  # upper bound for c
            # Use list comprehension to count valid triplets
            count += sum(1 for c in range(1, max_c + 1) if a * b + a * c + b * c <= n)

    return count
```

Figure 12: Step 50

Figure 13: Generated code samples by the algorithm

## A.4 ADAPTIVE SAMPLING SIZE

Gradient learning algorithms collect samples to train the surrogate gradient model. While more samples can potentially lead to more accurate models, it is important to curtail the number of samples in each training iteration to be able to execute enough optimization steps before consuming the budget. On the other hand, one advantage that gradient learning has over direct objective learning is that we train the model with pairs of samples instead of single evaluation points, s.t. for a sample set of size $n_s$ we can draw as many as $n_p = n_s^2$ pairs to train our model.

We empirically tested the optimal number of exploration size (i.e. $n_s$) in each training step and the optimal sample pair size ($n_p$) and found that a squared root profile is optimal both hyperparameters. Therefore, we use the following equation to control these parameters:

$$
\begin{aligned}
n_s &= 8 \cdot \left\lceil \sqrt{N} \right\rceil \\
n_p &= 2000 \cdot \left\lceil \sqrt{N} \right\rceil
\end{aligned}
\tag{9}
$$

where $N$ is the problem size.

## A.5 TRUST REGION MANAGEMENT

After finding the optimal solution inside the trust region, EGL shifts and scales down (i.e. shrinks) the trust region so that the optimal solution is centered at its origin. In case the decent path is long so that the minimum point is far away from the current trust region, this shrinking has the potential to slow down the learning rate and impede convergence. To prevent unnecessary shrinking of the trust region, we distinguish between two convergence types: **interior convergence**: in this case, we shrink the trust region while moving its center and **boundary convergence** where we only shift the center without applying shrinking (When the algorithm reaches the edges of the TR). This adjustment prevents fast convergence of the step size to zero before being able to sufficiently explore the input domain.

## A.6 Additional Empirical Results

Table 5: EGL Best Parameters

| Parameter | Value | Description |
|---|---|---|
| Exploration size | $8 \cdot \left\lceil \sqrt{\text{dimension}} \right\rceil$ | The number of iterations the algorithm will run. |
| Maximum movement to shrink | $0.2 \cdot \sqrt{\text{dimension}}$ | How much distance the algorithm needs to cover to prevent shrinking when reaching convergence. |
| Epsilon | $0.4 \cdot \sqrt{\text{dimension}}$ | Epsilon size to sample data. |
| Epsilon Factor | 0.97 | How much we shrink the epsilon each iteration. |
| Minimum epsilon | 0.0001 | The smallest epsilon we use. |
| Database size | $20,000 \cdot \left\lceil \sqrt{\text{dimension}} \right\rceil$ | How many tuples we used to train for each iteration. |
| Gradient network | dimension-10-15-10-dimension | What kind of network we use. |
| Budget | 150,000 | How much budget the algorithm uses. |

## A.7 Benchmark Algorithms

---
**Algorithm 3** CMA with a trust region

---
**Require:** $\texttt{total\_budget}, budget = 0, startpoint x, \delta = 1, \mu = 0, \gamma$
  **while** $budget \leq \texttt{total\_budget}$ **do**
    $\texttt{new\_generation}, \texttt{should\_stop} = CMA(x)$
    $\texttt{new\_generation} = mu + \delta * \texttt{new\_generation}$
    $evalutaions = space(\texttt{new\_generation})$
    $budget = budget + len(evalutaions)$
    $UpdateCovarMatrix(evaluations)$
    **if** $\texttt{should\_stop}$ **then**
      $\delta = \delta * \gamma$
      $\mu = argmin_{\texttt{x new\_generation}}(space(x))$
    **end if**
  **end while**

---

## A.8 CMA Versions

The trust region is key in the EGL algorithm (see Figures 1). We map between the search space ($\Omega$) and trust region ($\Omega^*$) using the tanh function, which provides smooth transitions but exhibits logarithmic behavior at the edges, slowing large steps. While helpful for problems with many local minima, this behavior restricts CMA's ability to take large steps, especially in high dimensions (see Figure 14). We developed two CMA variants: one with linear TR (L-CMA) and one with tanh TR(T-CMA). The linear TR allowed larger steps, improving exploration, while the tanh TR limited performance.

## A.9 Adversarial Attack Implementation

Our adversarial attack leverages the flexibility of black-box optimization (BBO), which is not constrained by differentiability requirements on the objective function. This allows us to manipulate the search space freely. Specifically, our attack focuses on modifying the positions of $m$ fixed-size boxes within the image, along with the permutations applied to

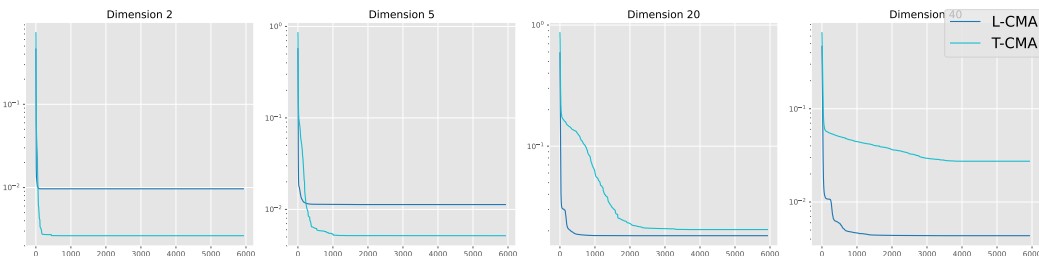

Figure 14: Comparing CMA with different trust regions types across different dimensions

the pixels inside these boxes. This approach provides two key advantages: it enables us to limit the extent of the alterations we introduce to the image and simultaneously reduces the dimensionality of the search space $\Omega$.

The penalty function we use balances two competing goals: minimizing the cross-entropy loss to ensure misclassification and limiting the perturbation magnitude using the mean squared error (MSE) loss. To achieve this balance, we define the following loss function:

$$s(ce) = \sigma\left(b \cdot (ce - \epsilon)\right)$$

$$Penalty_{mse}(ce, mse)\,(1 - s(ce)) \cdot mse^{n_1} + s(ce) \cdot (-mse^{n_2})$$

$$Penalty_{ce}(ce)\,(1 - s(ce))) \cdot e^{ce*n_1} + s(ce) \cdot (ln(ce^{n_2}))$$

$$Penalty(x) = Penalty_{mse}(ce(x), mse(x)) - Penalty_{ce}(ce(x))$$

In this formulation:

- mse $\in [0, 1]$ and ce $\in (-\infty, \infty)$.
- The terms $n_1 \leq n_2$ control the trade-off between the cross-entropy and MSE loss slopes.
- $\epsilon$ defines the minimum required cross-entropy loss to maintain misclassification.
- The parameter $b$ governs the rate of transition between minimizing cross-entropy and MSE losses.
- The function $\sigma$ is the sigmoid function, which controls how much focus is given to minimizing cross-entropy loss over MSE loss as ce increases.

This function balances the CR and the MSE, finding a minimum with CE to prevent detection by the classifier while minimizing the perturbation of the image.

## A.10 Full Robustness Experiment

Table 6: Comparison of Algorithm Versions on coco benchmark, comparing the error, the std and the budget it takes to finish 99% of the progress

| Version | Budget | Error | Std | Solved Problems |
|---------|--------|-------|-----|-----------------|
| **networks** | | | | |
| 20 40 20 | 58447.91 | 0.0060 | 0.0325 | 0.76 |
| 40 60 80 40 | 57960.44 | 0.0058 | 0.0325 | 0.76 |
| 60 80 60 | 58371.31 | 0.0060 | 0.0325 | 0.76 |
| 40 80 40 | 58159.34 | 0.0061 | 0.0325 | 0.76 |
| 40 15 10 40 | 58667.00 | 0.0061 | 0.0325 | 0.76 |
| **Epsilon** | | | | |
| 0.1 | 39491.81 | 0.006 | 0.0325 | 0.9 |
| | | | | Continued on next page |

| Version | Budget | Error | Std | Solved Problems |
|---|---|---|---|---|
| 0.4 | 48518.81 | 0.0057 | 0.0325 | 0.93 |
| 0.5 | 49469.81 | 0.0056 | 0.0325 | 0.93 |
| 0.6 | 54271.47 | 0.0061 | 0.0326 | 0.9 |
| 0.8 | 58667.00 | 0.0061 | 0.0325 | 0.9 |
| **Epsilon Factor** | | | | |
| 0.8 | 39275.72 | 0.0105 | 0.0365 | 0.83 |
| 0.9 | 47930.81 | 0.0068 | 0.0327 | 0.89 |
| 0.95 | 53899.78 | 0.0060 | 0.0326 | 0.9 |
| 0.97 | 58667.00 | 0.0061 | 0.0325 | 0.9 |
| 0.99 | 63676.47 | 0.0065 | 0.0326 | 0.89 |
| **Training Weights** | | | | |
| 50% | 58049.50 | 0.0059 | 0.0325 | 0.91 |
| 60% | 59477.47 | 0.0066 | 0.0333 | 0.89 |
| 70% | 57831.00 | 0.0057 | 0.0325 | 0.93 |
| 83% | 58667.00 | 0.0061 | 0.0325 | 0.9 |
| 100% | 59660.34 | 0.0061 | 0.0326 | 0.9 |
| **Perturbation** | | | | |
| 1 | 56807.97 | 0.0066 | 0.0333 | 0.89 |
| 0.3 | 58935.38 | 0.0088 | 0.0420 | 0.87 |
| 0.1 | 57699.78 | 0.0061 | 0.0325 | 0.89 |
| 0.01 | 58075.03 | 0.0060 | 0.0325 | 0.89 |
| 0 | 58667.00 | 0.0061 | 0.0325 | 0.89 |
| **Euclidean distance of the algorithm before shrinking trust region** | | | | |
| 0.4 | 57172.53 | 0.0061 | 0.0325 | 0.9 |
| 0.3 | 57324.53 | 0.0059 | 0.0325 | 0.91 |
| 0.2 | 58667.00 | 0.0057 | 0.0325 | 0.93 |
| 0.1 | 57327.50 | 0.0061 | 0.0325 | 0.89 |
| 0.01 | 57810.81 | 0.0062 | 0.0325 | 0.89 |
| **Value Normalizer LR** | | | | |
| 0.01 | 58211.59 | 0.0086 | 0.0420 | 0.83 |
| 0.05 | 58576.75 | 0.0061 | 0.0326 | 0.89 |
| 0.1 | 58667.00 | 0.0057 | 0.0325 | 0.93 |
| 0.2 | 59723.88 | 0.0059 | 0.0325 | 0.91 |
| 0.3 | 58194.97 | 0.0079 | 0.0370 | 0.87 |
| **Value Normalizer Outlier** | | | | |
| 0.01 | 57222.41 | 0.0059 | 0.0325 | 0.91 |
| 0.05 | 57776.38 | 0.0060 | 0.0325 | 0.9 |
| 0.1 | 58667.00 | 0.0057 | 0.0325 | 0.93 |
| 0.2 | 58481.75 | 0.0061 | 0.0325 | 0.9 |
| 0.3 | 58629.59 | 0.0063 | 0.0325 | 0.89 |
| **Trust Region Shrink Factor** | | | | |
| 0.99 | 103758.16 | 0.0177 | 0.0433 | 0.79 |
| 0.95 | 80660.69 | 0.0093 | 0.0346 | 0.81 |
| 0.9 | 58667.00 | 0.0057 | 0.0325 | 0.93 |
| 0.8 | 43503.22 | 0.0066 | 0.0325 | 0.87 |
| 0.7 | 39790.50 | 0.0075 | 0.0326 | 0.85 |
| **Optimizer LR** | | | | |
| 0.001 | 101550.59 | 0.0164 | 0.0364 | 0.76 |
| 0.005 | 64988.66 | 0.0066 | 0.0325 | 0.88 |
| 0.01 | 58667.00 | 0.0057 | 0.0325 | 0.93 |
| 0.02 | 62405.84 | 0.0056 | 0.0325 | 0.93 |
| | | | | Continued on next page |

| Version | Budget | Error | Std | Solved Problems |
|---------|--------|-------|-----|-----------------|
| 0.03 | 72041.81 | 0.0058 | 0.0325 | 0.93 |
| **Gradient LR** | | | | |
| 0.0001 | 63283.41 | 0.0063 | 0.0325 | 0.89 |
| 0.0005 | 59237.59 | 0.0086 | 0.0420 | 0.84 |
| 0.001 | 58667.00 | 0.0057 | 0.0325 | 0.93 |
| 0.002 | 58776.25 | 0.0061 | 0.0325 | 0.9 |
| 0.003 | 57934.91 | 0.0061 | 0.0326 | 0.9 |

## A.11 Theoretical Analysis

### A.11.1 Mean Gradient Accuracy

**Definition 1** *The second-order mean gradient around $x$ of radius $\epsilon > 0$:*

$$\tilde{g}_\varepsilon(x) = \arg\min_{g \in \mathbb{R}^n} \int_{V_\varepsilon(x)} \left| g^\top \tau + \tfrac{1}{2}\tau^\top J(g)\tau - [f(x+\tau) - f(x)] \right|^2 d\tau.$$

**Theorem 3** *(Improved Controllable Accuracy): For any twice differentiable function $f \in \mathcal{C}^2$, there exists $\kappa_g(x) > 0$ such that for any $\varepsilon > 0$, the second-order mean-gradient $\tilde{g}_\varepsilon(x)$ satisfies*

$$\|\tilde{g}_\varepsilon(x) - \nabla f(x)\| \le \kappa_g(x)\varepsilon^2 \quad \text{for all } x \in \Omega.$$

*Proof.* Since $f \in \mathcal{C}^2$, the Taylor expansion around $x$ is

$$f(x+\tau) = f(x) + \nabla f(x)^\top \tau + \tfrac{1}{2}\tau^\top H(x)\tau + R_x(\tau),$$

where $H(x)$ is the Hessian matrix at $x$, and the remainder $R_x(\tau)$ satisfies

$$|R_x(\tau)| \le \tfrac{1}{6}k_g\|\tau\|^3,$$

By the Definition of $g_\epsilon$, an upper bound of $\mathcal{L}(g_\epsilon(x))$ is:

$$\mathcal{L}(g_\epsilon(x)) \le \mathcal{L}(\nabla f(x)) = \int_{V_\varepsilon(x)} \left| \nabla f(x)^\top \tau + \tfrac{1}{2}\tau^\top H(x)\tau - \left(f(x+\tau) - f(x)\right) \right|^2 d\tau = \int_{V_\varepsilon(x)} |R_x(\tau)|^2 d\tau$$

$$\le \left(\tfrac{1}{6}k_g\right)^2 \int_{V_\varepsilon(x)} \|\tau\|^6 d\tau = \left(\tfrac{1}{6}k_g\right)^2 |V_1(x)|\varepsilon^{n+6}.$$

We now find a lower bound:

lets define for convenience $\delta_g = (g_\varepsilon(x) - \nabla f(x))$, $\delta_H = J(g_\varepsilon(x)) - H(x)$

$$\mathcal{L}(g_\varepsilon(x)) = \int_{V_\varepsilon(x)} |g_\varepsilon(x)\tau - \nabla f(x)\tau + \nabla f(x)\tau + \tfrac{1}{2}\tau J(g_\varepsilon(x))\tau -$$

$$\tfrac{1}{2}\tau H(x)\tau + \tfrac{1}{2}\tau H(x)\tau - f(x+\tau) + f(x)|^2 d\tau$$

$$= \int_{V_\varepsilon(x)} (\delta_g^\top \tau + \tfrac{1}{2}\tau^\top \delta_H \tau - R_x(\tau))^2 d\tau$$

$$= \int_{V_\varepsilon(x)} ((\delta_g^\top \tau)^2 + \tfrac{1}{4}(\tau^\top \delta_H \tau)^2 + R_x(\tau)^2 - 2\delta_g^\top \tau R_x(\tau) - \tau^\top \delta_H \tau R_x(\tau) + \tau^\top \delta_g \tau^\top \delta_H \tau) d\tau$$

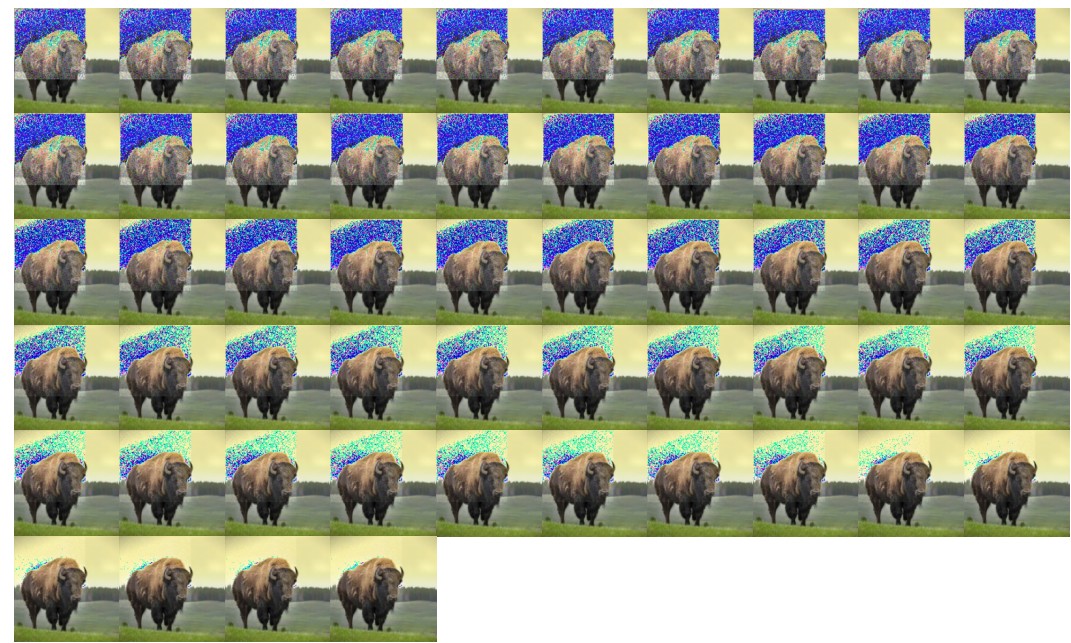

Figure 15: The search trajectory for an adversarial image generation with OGL

Since $a^2 + b^2 \geq 2ab$ we conclude that $\frac{1}{4}(\tau^\top \delta_H \tau)^2 + R_x(\tau)^2 - \tau^\top \delta_H \tau R_x(\tau) \geq 0$

$$\mathcal{L}(g_\varepsilon(x)) = \int_{V_\varepsilon(x)} ((\delta_g^\top \tau)^2 + \frac{1}{4}(\tau^\top \delta_H \tau)^2 + R_x(\tau)^2 - 2\delta_g^\top \tau R_x(\tau) - \tau^\top \delta_H \tau R_x(\tau) + \tau^\top \delta_g \tau^\top \delta_H \tau) d\tau$$

$$\geq \int_{V_\varepsilon(x)} ((\delta_g^\top \tau)^2 - 2\delta_g^\top \tau R_x(\tau) + \tau^\top \delta_g \tau^\top \delta_H \tau) d\tau$$

$$= \int_{V_\varepsilon(x)} (\delta_g^\top \tau)^2 d\tau - 2 \int_{V_\varepsilon(x)} \delta_g^\top \tau R_x(\tau) d\tau + \int_{V_\varepsilon(x)} \tau^\top \delta_g \tau^\top \delta_H \tau d\tau$$

$$= \int_{V_\varepsilon(x)} (\delta_g^\top \tau)^2 d\tau - 2 \int_{V_\varepsilon(x)} \delta_g^\top \tau R_x(\tau) d\tau + \int_{V_\varepsilon(x)} \tau^\top \delta_g \tau^\top \delta_H \tau d\tau$$

We will split the equation into 3 components: First $A = \int_{V_\varepsilon(x)} (\delta_g^\top \tau)^2 d\tau$ , is we open this expression we see that

$$(\delta_g^\top \tau)^2 = \sum_{i=1}^n \sum_{j=1}^n (\delta_g)_i (\delta_g)_j \tau_i \tau_j.$$

Since $V_\varepsilon(x)$ is symmetric around $x$ odd moments of $\tau$ integrate to zero, therefore

$$A = \int_{V_\varepsilon(x)} (\delta_g^\top \tau)^2 d\tau = \int_{V_\varepsilon(x)} (\sum_{i=1}^n \sum_{j=1}^n (\delta_g)_i (\delta_g)_j \tau_i \tau_j) d\tau = \int_{V_\varepsilon(x)} (\sum_{i=1}^n (\delta_g)_i^2 \tau_i^2) d\tau = ||\delta_g||^2 \varepsilon^{n+2} |V_1(x)|$$

Let $B = 2 \int_{V_\varepsilon(x)} \delta_g^\top \tau R_x(\tau) d\tau$

$$B = 2 \int_{V_\varepsilon(x)} \delta_g^\top \tau R_x(\tau) d\tau \leq 2 \int_{V_\varepsilon(x)} ||\delta_g|| \cdot ||\tau|| \cdot ||R_x(\tau)|| d\tau \leq 2||\delta_g|| \int_{V_\varepsilon(x)} ||\tau|| \frac{1}{6} k_g \tau^3 d\tau$$

$$\leq \frac{1}{3} k_g ||\delta_g|| \int_{V_\varepsilon(x)} ||\tau||^4 d\tau = \frac{1}{3} k_g ||\delta_g|| \varepsilon^{n+4} |V_1(x)|$$

Finally $C = \int_{V_\varepsilon(x)} \tau^\top \delta_g \tau^\top \delta_H \tau d\tau$, we should remember the property derived from $V_\varepsilon(x)$ symmetry. If we open the vector multiplication we get

$$\tau^\top \delta_g \tau^\top \delta_H \tau = \sum_{i=1}^{n}\sum_{j=1}^{n}\sum_{k=1}^{n} \tau_i (\delta_g)_i (\delta_H)_{jk} \tau_j \tau_k = \sum_{i=1}^{n}\sum_{j=1}^{n}\sum_{k=1}^{n} (\delta_g)_i (\delta_H)_{jk} \tau_i \tau_j \tau_k.$$

Here there is no way for an even moment of $\tau$ to exist so $C = 0$ To sum this up, we get

$$\mathcal{L}(g_\varepsilon(x)) \geq A - B + C \geq ||\delta_g||^2 \varepsilon^{n+2} |V_1(x)| - \tfrac{1}{3} k_g ||\delta_g|| \varepsilon^{n+4} |V_1(x)|$$

We combine the lower and upper-bound

$$||\delta_g||^2 \varepsilon^{n+2} |V_1(x)| - \tfrac{1}{3} k_g ||\delta_g|| \varepsilon^{n+4} |V_1(x)| \leq (\tfrac{1}{6} k_g)^2 |V_1(x)| \varepsilon^{n+6}$$

$$||\delta_g||^2 - \tfrac{1}{3} k_g ||\delta_g|| \varepsilon^2 - (\tfrac{1}{6} k_g)^2 \varepsilon^4 \leq 0$$

$$||\delta_g||^2 \leq \frac{\tfrac{1}{3} k_g \varepsilon^2 + \sqrt{\tfrac{1}{9} k_g^2 \varepsilon^4 + 4(\tfrac{1}{6} k_g)^2 \varepsilon^4}}{2} = \frac{\tfrac{1}{3} k_g \varepsilon^2 + \sqrt{2 \tfrac{1}{9} k_g^2 \varepsilon^4}}{2} = \tfrac{1}{3\sqrt{2}} k_g \varepsilon^2 = k_g(x) \varepsilon^2$$

### A.11.2 CONVERGENCE ANALYSIS

**Theorem 4** *Let $f : \Omega \to \mathbb{R}$ be a convex function with Lipschitz continuous gradient, i.e. $f \in \mathcal{C}^{+1}$ and a Lipschitz constant $\kappa_f$ and let $f(x^*)$ be its optimal value. Suppose a controllable mean-gradient model $g_\varepsilon$ with error constant $\kappa_g$, the gradient descent iteration $x_{k+1} = x_k - \alpha g_\varepsilon(x_k)$ with a sufficiently small $\alpha$ s.t. $\alpha \leq \min(\frac{1}{\kappa_g}, \frac{1}{\kappa_f})$ guarantees:*

1. *For $\varepsilon \leq \frac{||\nabla f(x)||}{5\alpha}$, monotonically decreasing steps s.t. $f(x_{k+1}) \leq f(x_k) - 2.25\frac{\varepsilon^2}{\alpha}$.*

2. *After a finite number of iterations, the descent process yields $x^\star$ s.t. $||\nabla f(x^\star)|| \leq \frac{5\varepsilon}{\alpha}$.*

*Proof.* For a convex function with Lipschitz continuous gradient, the following inequality holds for all $x_k$

$$f(x) \leq f(x_k) + (x - x_k) \cdot \nabla f(x_k) + \frac{1}{2} \kappa_f ||x - x_k||^2 \tag{10}$$

Plugging in the iteration update $x_{k+1} = x_k - \alpha g_\varepsilon(x_k)$ we get

$$f(x_{k+1}) \leq f(x_k) - \alpha g_\varepsilon(x_k) \cdot \nabla f(x_k) + \alpha^2 \frac{1}{2} \kappa_f ||g_\varepsilon(x_k)||^2 \tag{11}$$

For a controllable mean-gradient we can write $||g_\varepsilon(x) - \nabla f(x)|| \leq \varepsilon^2 \kappa_g$, therefore we can write $g_\varepsilon(x) = \nabla f(x) + \varepsilon^2 \kappa_g \xi(x)$ s.t. $||\xi(x)|| \leq 1$ so the inequality is

$$f(x_{k+1}) \leq f(x_k) - \alpha||\nabla f(x_k)||^2 - \alpha\varepsilon^2 \kappa_g \xi(x_k) \cdot \nabla f(x_k) + \alpha^2 \frac{1}{2} \kappa_f ||\nabla f(x) + \varepsilon^2 \kappa_g \xi(x)||^2 \tag{12}$$

Using the equality $||a + b||^2 = ||a||^2 + 2a \cdot b + ||b||^2$ and the Cauchy-Schwartz inequality inequality $a \cdot b \leq ||a|| ||b||$ we can write

$$f(x_{k+1}) \leq f(x_k) - \alpha||\nabla f(x_k)||^2 - \alpha\varepsilon^2 \kappa_g ||\xi(x_k)|| \cdot ||\nabla f(x_k)||$$

$$+ \alpha^2 \frac{1}{2} \kappa_f \left( ||\nabla f(x)||^2 + 2\varepsilon^2 \kappa_g ||\xi(x)|| \cdot ||\nabla f(x_k)|| + \varepsilon^4 \kappa_g^2 ||\xi(x)||^2 \right)$$

$$\leq f(x_k) - \alpha||\nabla f(x_k)||^2 - \alpha\varepsilon^2 \kappa_g ||\nabla f(x_k)|| \tag{13}$$

$$+ \frac{\alpha^2}{2} \kappa_f ||\nabla f(x)||^2 + \alpha^2 \kappa_f \varepsilon^2 \kappa_g ||\nabla f(x_k)|| + \frac{\alpha^2 \varepsilon^4}{2} \kappa_f \kappa_g^2$$

$$= f(x_k) - (\alpha - \frac{k_f}{2}\alpha^2)||\nabla f(x_k)||^2 + (-\alpha + k_f\alpha^2)||\nabla f(x_k)||\varepsilon^2 \kappa_g + \frac{k_f}{2}\alpha^2\varepsilon^4 \kappa_g^2$$

Using the requirement $\alpha \leq \min(\frac{1}{\kappa_g}, \frac{1}{\kappa_f})$ it follows that $\alpha\kappa_g \leq 1$ and $\alpha\kappa_f \leq 1$ so

$$f(x_{k+1}) \leq f(x_k) - (\alpha - \frac{k_f}{2}\alpha^2)||\nabla f(x_k)||^2 + (-\alpha + k_f\alpha^2)||\nabla f(x_k)||\varepsilon^2 \kappa_g + \frac{k_f}{2}\alpha^2\varepsilon^4 \kappa_g^2$$

$$\tag{14}$$

$$= f(x_k) - \frac{\alpha}{2}||\nabla f(x_k)||^2 + \frac{k_f}{2}\alpha^2\varepsilon^4 \kappa_g^2 \leq f(x_k) - \frac{\alpha}{2}||\nabla f(x_k)||^2 + \frac{\varepsilon^4}{2}\kappa_g$$

Now, for $x$ s.t. $\|\nabla f(x)\| \geq \frac{5\varepsilon^2}{\alpha}$ then $\varepsilon^2 \leq \|\nabla f(x)\|\frac{\alpha}{5}$. Plugging it into our inequality we obtain

$$
\begin{aligned}
f(x_{k+1}) &\leq f(x_k) - \frac{\alpha}{2}\|\nabla f(x_k)\|^2 + \frac{\alpha^2}{50}\kappa_g\|\nabla f(x_k)\|^2 \\
&\leq f(x_k) - \frac{\alpha}{2}\|\nabla f(x_k)\|^2 + \frac{\alpha}{50}\|\nabla f(x_k)\|^2 \\
&= f(x_k) - 0.48\alpha\|\nabla f(x_k)\|^2 \\
&\leq f(x_k) - 12\frac{\varepsilon^4}{\alpha}
\end{aligned}
\tag{15}
$$

Therefore, for all $x$ s.t., $\|\nabla f(x)\| \geq \frac{5\varepsilon^2}{\alpha}$ we have a monotonically decreasing step with finite size improvement, after a finite number of steps we obtain $x^\star$ for which $\|\nabla f(x^\star)\| \leq \frac{5\varepsilon^2}{\alpha}$.

### A.11.3 OPTIMISTIC GRADIENT ACCURACY

**Definition 2** *The optimistic gradient around $x$ of radius $\epsilon > 0$:*

$$
\tilde{g}_\varepsilon(x) = \arg\min_{g\in\mathbb{R}^n} \int_{V_\varepsilon(x)} w(\tau)\left|g^\top\tau - [f(x+\tau)-f(x)]\right|^2 d\tau.
$$

**Theorem 5** *(Optimistic gradient controllable Accuracy): For any twice differentiable function $f \in \mathcal{C}^2$, there exists $\kappa_g(x) > 0$ such that for any $\varepsilon > 0$, the second-order mean-gradient $\tilde{g}_\varepsilon(x)$ satisfies*

$$
\|\tilde{g}_\varepsilon(x) - \nabla f(x)\| \leq \kappa_g(x)\varepsilon \quad \text{for all } x \in \Omega.
$$

*Proof.* Since $f \in \mathcal{C}^2$ the Taylor expansion around $x$ is

$$
f(x+\tau) = f(x) + \nabla f(x)^\top\tau + R_x(\tau),
$$

where $H(x)$ is the Hessian matrix at $x$, and the remainder $R_x(\tau)$ satisfies

$$
|R_x(\tau)| \leq \tfrac{1}{2}k_g\|\tau\|^2,
$$

By definition $g_\epsilon$, an upper bound $\mathcal{L}(g_\epsilon(x))$ is:

$$
\begin{aligned}
\mathcal{L}(g_\epsilon(x)) &\leq \mathcal{L}(\nabla f(x)) = \int_{V_\varepsilon(x)} \left|\nabla f(x)^\top\tau - \left(f(x+\tau)-f(x)\right)\right|^2 d\tau = \int_{V_\varepsilon(x)} |R_x(\tau)|^2 d\tau \\
&\leq \left(\tfrac{1}{2}k_g\right)^2 \int_{V_\varepsilon(x)} \|\tau\|^4 d\tau = \left(\tfrac{1}{2}k_g\right)^2 |V_1(x)|\varepsilon^{n+4}.
\end{aligned}
$$

*Proof.* The optimistic mean gradient around x of radius $\epsilon > 0$:

$$
\tilde{g}_\varepsilon(x) = \arg\min_{g\in\mathbb{R}^n} \int_{V_\varepsilon(x)} w(\tau)\left|g^\top\tau - [f(x+\tau)-f(x)]\right|^2 d\tau.
$$

$$
\begin{aligned}
\mathcal{L}(g_\varepsilon(x)) &= \int_{V_\varepsilon(x)} w(\tau)(\delta_g^\top\tau - R_x(\tau))^2 d\tau \geq \int_{V_\varepsilon(x)} w(\tau)(\delta_g^\top\tau)^2 - 2\cdot w(\tau)\delta_g^\top R_x(\tau) + w(\tau)R_x(\tau)^2 d\tau \\
&= \int_{V_\varepsilon(x)} w(\tau)(\delta_g^\top\tau)^2 - \int_{V_\varepsilon(x)} 2\cdot w(\tau)\delta_g^\top \cdot R_x(\tau)d\tau + \int_{V_\varepsilon(x)} w(\tau)R_x(\tau)^2 d\tau \\
&\geq \int_{V_\varepsilon(x)} w(\tau)(\delta_g^\top\tau)^2 - 2\int_{V_\varepsilon(x)} w(\tau)\delta_g^\top \cdot R_x(\tau)d\tau
\end{aligned}
$$

Since $V_\varepsilon(x)$ is symmetric around $x$ odd moments of $\tau$ integrate to zero, therefore

$$
\begin{aligned}
A &= \int_{V_\varepsilon(x)} w(\tau)(\delta_g^\top\tau)^2 d\tau = \int_{V_\varepsilon(x)} w(\tau)(\sum_{i=1}^{n}\sum_{j=1}^{n}(\delta_g)_i(\delta_g)_j\tau_i\tau_j)d\tau \\
&= \int_{V_\varepsilon(x)} w(\tau)(\sum_{i=1}^{n}(\delta_g)_i^2\tau_i^2)d\tau = \int_{V_\varepsilon(x)} w(\tau)\|\delta_g\|^2 \cdot \|\tau\|^2 d\tau \\
&= \|\delta_g\|^2\varepsilon^{n+2}|V_1(x)|\int_{V_\varepsilon(x)} w(\tau) \leq \|\delta_g\|^2\varepsilon^{n+2}|V_1(x)|W_u
\end{aligned}
$$

Let $B = 2 \int_{V_\varepsilon(x)} w(\tau) \delta_g^\top \tau R_x(\tau) d\tau$

$$B = 2 \int_{V_\varepsilon(x)} w(\tau) \delta_g^\top \tau R_x(\tau) d\tau \leq 2 \int_{V_\varepsilon(x)} w(\tau) \|\delta_g\| \cdot \|\tau\| \cdot R_x(\tau) d\tau \leq 2\|\delta_g\| \int_{V_\varepsilon(x)} w(\tau) \|\tau\| \tfrac{1}{2} k_g \tau^2 d\tau$$

$$\leq \tfrac{1}{3} k_g \|\delta_g\| \int_{V_\varepsilon(x)} w(\tau) \|\tau\|^3 d\tau = k_g \|\delta_g\| \varepsilon^{n+3} |V_1(x)| W_u$$

To sum this up, we get

$$\mathcal{L}(g_\varepsilon(x)) \geq A - B \geq \|\delta_g\|^2 \varepsilon^{n+2} |V_1(x)| W_u - k_g \|\delta_g\| \varepsilon^{n+3} |V_1(x)| W_u$$

We combine the lower and upper-bound

$$\|\delta_g\|^2 \varepsilon^{n+2} |V_1(x)| W_u - k_g \|\delta_g\| \varepsilon^{n+3} |V_1(x)| W_u \leq (\tfrac{1}{2} k_g)^2 |V_1(x)| \varepsilon^{n+4}$$

$$\|\delta_g\|^2 W_u - k_g \|\delta_g\| \varepsilon W_u - (\tfrac{1}{2} k_g)^2 \varepsilon^2 \leq 0$$

$$\|\delta_g\| \leq \frac{k_g \varepsilon W_u + \sqrt{k_g^2 \varepsilon^2 W_u^2 + 4 W_u (\tfrac{1}{2} k_g)^2 \varepsilon^2}}{2 W_u} = k_g \varepsilon \frac{W_u + \sqrt{W_u^2 + W_u}}{2 W_u}$$

