# OpenReview forum: "Optimistic Gradient Learning with Hessian Corrections for High-Dimensional Black-Box Optimization"
_ICLR.cc/2025/Conference — Submitted to ICLR 2025_

### Official Review · Reviewer_3M45 · 2024-10-23

**Soundness:** 3
**Presentation:** 3
**Contribution:** 3
**Rating:** 6
**Confidence:** 2

**Summary:**

This paper extends the gradient descent learning method by incorporating second-order Taylor corrections through the function's Hessian, enhancing robustness in complex, highly non-linear problems. The OGL, HGL, and OHGL algorithms are presented and validated on the synthetic COCO suite, as well as in two scenarios: adversarial attacks and code generation.

**Strengths:**

1. The idea of Optimistic Gradient Learning and Higher-Order Gradient Learning is novel.
2. These new algorithms have achieved better results in experiments.

**Weaknesses:**

1. The reference format is not uniform. I suggest the authors improve them.
2. There are some mistakes in this paper, for example, the CE loss in the title of Table 3 is not reflected in the table.
3. It seems that the most recommended algorithm in this paper, OHGL, has not been applied in real-world scenarios.

**Questions:**

1. I think the experiments of HGL and OHGL in adversarial attacks and code generations can further enhance persuasiveness.
2. Format modification is necessary, reference and appendix are confusing.

---

> ### Author Response · Authors · 2024-11-18
> **Response to Reviewer 3M45**
>
> Thank you for your feedback and suggestions:
>
> •	Regarding Weakness 1+2: Thank you for pointing this out. We have corrected the reference page and addressed other mistakes, including the CE loss issue in Table 3. A revised version of the paper will be uploaded.
>
> •	Regarding Question 1: Following your comment, we are conducting additional experiments with the HGL+OHGL algorithm on the adversarial attack problem to further support our findings.

---

> ### Author Response · Authors · 2024-11-28
> **Response for Reviewer 3M45**
>
> We are truly grateful for your review. Your comments have been instrumental in improving the clarity and quality of our paper, allowing us to address potential misunderstandings and refine our approach.
>
> In this revised version, we have fixed all the errors we could identify and fix the error on the reference page. Additionally, we conducted further experiments to demonstrate the value of each component of our methodology (see Figure 4(b) on page 7). We also applied HGL to the adversarial attack problem, showcasing its effectiveness in addressing real-world challenges.
>
> We hope these enhancements address the concerns you raised. We deeply appreciate the time and effort you have invested in reviewing our manuscript, and we sincerely hope the revised version addresses any concerns you have. Thank you once again for your feedback.

---

### Official Review · Reviewer_eUep · 2024-11-01

**Soundness:** 3
**Presentation:** 2
**Contribution:** 2
**Rating:** 5
**Confidence:** 3

**Summary:**

This work proposes an OHGL framework to solve black-box optimization. It improves the previous EGL framework by incorporating second-order Taylor corrections via the function’s Hessian and assigning higher weights to samples with lower objective values. Experiments on adversarial attack and code generation show its potential in high-dimension real-world problems.

**Strengths:**

1. The proposed Hessian corrections and weighed gradient are well-motivated, effectively addressing the limitations of the previous EGL framework.
2. Real-world high-dimension examples, including adversarial attacks and code generation, demonstrate the scalability and efficiency of this framework.

**Weaknesses:**

1. Compared to previous work, such as EGL or model-based methods [1], the theoretical understanding of the proposed Hessian corrections and weighted gradients is lacking. A more rigorous theoretical analysis would help justify the design choices

2. The work introduces several algorithm-level designs in Sec. 5, such as adaptive sampling size and trust-region management. The tolerance analysis in Section 6.2 suggests that the influence of these designs can accumulate over time, potentially leading to significant changes in performance. However, the paper does not provide clear guidance on how to tune these parameters for different black-box optimization tasks with a limited budget.

[1] Kumar, A., & Levine, S. (2020). Model inversion networks for model-based optimization. Advances in neural information processing systems, 33, 5126-5137.

**Questions:**

1. The OHGL framework incorporates Hessian information during the training stage. However, it is unclear why the second-order information is not utilized when updating the next point (line 278).
2. Is the hyperparameter tuning process conducted for all baselines?
3. Which design in the OHGL framework contributes the most to the observed performance improvements?

---

> ### Author Response · Authors · 2024-11-18
> **Response to Reviewer eUep**
>
> Thank you for your thoughtful feedback.
>
> Regarding Weakness 1: Thank you for highlighting this. In our revision, we added a theoretical analysis of the convergence of WGL to the true gradient as well as the convergence of the learning process to a local minima. We believe that these results will strengthen the understanding of our algorithms as well as some of our design choices.
>
> Regarding Weakness 2: In the hyperparameter tolerance section, we discuss the algorithm's resilience to most parameters, such as training dataset size and sampling size. However, as you noted correctly, certain parameters (e.g., epsilon factor, shrink factor, and outlier learning rate) do have cumulative effects. EGL's paper provides guidelines for stable convergence regarding epsilon and shrink factors.
> Regarding Question 1: You raise an excellent point. We experimented with using approximated Hessian information for calculating the next point (e.g., Newton's methods). However, these approaches often hindered algorithm progress. The Adam optimizer, with its established benefits, consistently showed superior results.
> Regarding Question 2: For our gradient learning algorithms, we used the default parameters as in the EGL paper, except for the adaptive sampling size which we described in the “Fine-tuning the gradient learning algorithm” section. For CMA we also used the default Hyperparameters without further tuning.
>
> Regarding Question 3: Following your comment, we conducted an ablation study. The results confirm that the higher-order Taylor approximation and the optimistic method are the primary contributors to performance improvements. The optimistic method handles non-retractable environments effectively, while second-order Taylor corrections enhance gradient accuracy in stable environments.
>
> We hope our revision will satisfy your raised concerns.

---

> ### Author Response · Authors · 2024-11-28
> **Response for Reviewer eUep Regarding the revised paper**
>
> We would like to express our sincere gratitude for your detailed feedback on our paper. Your insights have been instrumental in identifying areas where our work could be strengthened, and we believe that the revisions we’ve made significantly improve the clarity and rigor of our approach.
>
> In the revised version of the paper, we have addressed the theoretical background for HGL and OGL, which was underexplored in the previous version. Specifically, we have included proofs of the convergence rate of the algorithm and the accuracy of the mean gradient estimation. These additions can be found in the appendix (page 20) and the corresponding theoretical sections (Sections 3 and 4). Furthermore, this theoretical analysis has been leveraged to provide practical guidance on selecting appropriate hyper-parameters (page 8, Section 6.2) and other key decisions.
>
> We have also elaborated on key decisions that previously raised concerns, such as the use of the Hessian (see the footnote on page 5). Additionally, we conducted an ablation study to evaluate the contributions of each component of our method. Figure 4(b) (page 7) highlights the value added by the optimistic and Hessian correction strategies, which we hope effectively demonstrate their importance in our framework.
>
> We hope these revisions address your concerns and provide a more comprehensive and polished presentation of our work. Your expertise is greatly valued, and we are grateful for the time and effort you have dedicated to reviewing our manuscript. We look forward to your thoughts on this updated version and hope it now meets the standards expected.

---

### Official Review · Reviewer_86Mz · 2024-11-04

**Soundness:** 2
**Presentation:** 2
**Contribution:** 2
**Rating:** 5
**Confidence:** 2

**Summary:**

This paper proposes Optimistic Higher-Order Gradient Learning (OHGL), using second-order Taylor approximation, for solving black-box optimization (BBO) problems. Furthermore, the authors refine EGL's sampling strategy and loss function to improve its efficiency in high-dimensional settings. The proposed OHGL method is validated through experiments on the synthetic COCO suite and applied to two real-world machine learning tasks: adversarial training and code generation. The experimental results indicate that OHGL generates stronger candidate solutions, positioning it as a valuable tool for machine learning researchers and practitioners addressing high-dimensional, non-linear optimization challenges.

**Strengths:**

1. The concept of enhancing Explicit Gradient Learning (EGL) with second-order corrections is both interesting and novel, to the best of my knowledge.
2. The paper is generally well-written and presents its ideas in a clear and accessible manner.

**Weaknesses:**

1. My primary concern relates to the computational cost associated with Hessian computation or approximation, which may hinder the scalability of the proposed method. Given that memory- and computation-efficient Hessian approximation techniques are well-established in the literature, I strongly recommend that the authors evaluate their proposed method using more efficient approximations at larger scales.

2. The enhancements over EGL are attained through the integration of several design elements; however, a thorough ablation study is notably absent from the paper.

**Questions:**

See weaknesses.

---

> ### Author Response · Authors · 2024-11-18
> **Response for Reviewer 86Mz**
>
> Thank you, we appreciate your valuable feedback and concerns,
>
> •	Regarding Weakness 1: Following your suggestion, we tested approximation methods such as BFGS and K-FAC. However, many of these methods were not differentiable, preventing back-propagation through the calculations. Others were either inaccurate or unstable, failing to improve accuracy during training. Importantly, we found that back-propagating on the Hessian is unnecessary for improving EGL results. Using the Hessian only to minimize the Taylor error avoids the need for additional back-propagation steps, keeping complexity in the same order of magnitude.
>
> •	Regarding Weakness 2: Thank you for pointing this out. We conducted an ablation study to assess the impact of each element in our method. The results show that the optimistic method and second-order Taylor expansion are the most significant contributors. Using both enables the algorithm to solve the highest number of problems in the COCO benchmark.

---

> > ### Comment · Reviewer_86Mz · 2024-11-26
> >
> > Thank you for your response.
> >
> > I believe that backpropagation through the calculation of the Hessian is not practical due to the significant additional computational overhead, especially in large-scale scenarios. Blocking backpropagation through the Hessian seems like a natural workaround to reduce computation, and I am curious to know whether a similar performance improvement can be achieved as in OEGL. Additionally, it would be helpful to understand whether the conclusions of the ablation study remain unchanged under this modified approach.

---

> > > ### Author Response · Authors · 2024-11-27
> > > **Response for Reviewer 86Mz**
> > >
> > > You are straight to the point, following your comments, we've added the modified version of HGL to the ablation study.
> > > You can see it in the modified version of the paper in Figure 4(b) (page 7 if nothing changes)
> > > The results indicate that using HGL still benefits the optimization.
> > >
> > > Thank you for your notes.

---

> ### Author Response · Authors · 2024-11-28
> **Response for Reviewer 86Mz**
>
> We deeply appreciate your thoughtful and constructive review. Your feedback has been instrumental in helping us enhance the clarity and rigor of our paper, enabling us to better articulate our contributions and address key challenges in our methodology.
>
> Following your advice, we explored multiple methods for calculating the Hessian in the Hessian correction method. Our experiments revealed that using the detached Hessian yields virtually identical results while significantly reducing computational costs. Additionally, To further substantiate the impact of our modifications, we have included an ablation study in the revised manuscript, clearly demonstrating the value of each adjustment described in the paper, you can find it in Figure 4(b) (page 7) in the revised version.
>
> We are confident that these improvements address the primary concerns you raised and hope the updated version presents a clearer and more compelling representation of our work. Thank you once again for your invaluable insights and for considering the revised manuscript. Your expertise and time are greatly appreciated.

---

### Official Review · Reviewer_YVhg · 2024-11-09

**Soundness:** 3
**Presentation:** 3
**Contribution:** 3
**Rating:** 6
**Confidence:** 4

**Summary:**

The paper introduces an enhanced method for black-box optimization (BBO) by specifically addressing challenges in high-dimensional and complex optimization landscapes. The proposed approach, termed Optimistic Higher-Order Gradient Learning (OHGL), builds on previous methods like Explicit Gradient Learning (EGL) by incorporating both optimistic gradient estimation and Hessian corrections. By leveraging second-order Taylor expansions and Hessian information, the method improves gradient accuracy. The authors propose a sampling profiler to reduce required samples and refine trust region adjustments for enhanced convergence efficiency. It demonstrates strong potential in machine learning tasks where gradient information is inaccessible or costly to compute.

**Strengths:**

The paper appears to be well-grounded in both theoretical and experimental methodologies. The use of both first- and second-order gradient information for better accuracy in high-dimensional spaces is a technically sound choice, demonstrating an understanding of gradient approximation complexities in BBO.

The breakdown of components like Optimistic Gradient Learning and Higher-Order Gradient Learning helps clarify how each extension builds on EGL, and the inclusion of figures and empirical results assists in understanding performance benefits. However, a possible point for improvement might be adding more intuitive explanations or visual aids for readers less familiar with gradient learning concepts.

**Weaknesses:**

1. Computational Complexity and Efficiency:
The inclusion of Hessian corrections, while beneficial for accuracy, introduces substantial computational cost, particularly in high-dimensional spaces.

2. Generalization of Results:
The results focus heavily on the COCO test suite and two specific applications. However, broader generalizability to other complex, black-box settings, such as reinforcement learning or sequential decision-making tasks, remains unaddressed.

3. Trust Region's Limitations:
The algorithm's handling of boundary conditions may slow down convergence in real-world, high-dimensional applications due to excessive shrinking, which could be inefficient in practice.

**Questions:**

1. The inclusion of comparisons to more recent methods like Bayesian optimization frameworks or evolutionary algorithms would strengthen the claims regarding OHGL’s effectiveness.

2. The organization could be refined for flow, as several sections (like trust region management and adaptive sampling) feel slightly disjointed.

---

> ### Author Response · Authors · 2024-11-18
> **Response for Reviewer YVhg**
>
> Thank you for your feedback and questions which we address below:
>
> •	Regarding Weakness 1: Following your comment, we conducted additional experiments and found that calculating the Hessian without propagating through it achieves almost the same results without significantly increasing complexity. Since back-propagation for training takes O(n), calculating the Hessian adds a similar order of complexity, resulting in O(2n), which remains manageable (compared to O(n^2) which is the cause of back-propagating through the Hessian calculation). This adjustment keeps CPU time and memory overhead negligible.
>
> •	Regarding Weakness 2: While akin to BBO, Reinforcement learning (RL) and sequential decision-making tasks bear several different characteristics which make the generalization of EGL to RL not trivial. First, the common q-value function in sequential decision problems is a linear summation of instantaneous rewards which can be reconstructed (and learned) using Temporal Difference learning and the Bellman operator.
> This means that to evaluate the q-value function (the equivalent of f() in BBO) and reconstruct the mean gradient, we need to have a model for the q-value function of the next state. This requires learning a model for the objective (as done in classic RL) and not just for the gradient, as demonstrated in our work. Next, RL requires a state-conditioned policy (unlike classic optimization problem, in which the optimal action x* is not conditioned on the state), this means that we do not necessarily collect two samples in the same state (s, a1) and (s, a2) which are required to evaluate the action gradient. Lastly, RL generally considers stochasticity in the environment which may add another layer of complexity in the analysis. While we have already invested in generalizing our method to the RL setting, and obtained some positive initial results, focusing on the classic case of a stateless problem with a single reward which finds application across many ML problems, has merit on its own.
>
> •	Regarding Weakness 3: We also identified this issue during our research and addressed it in the "Trust Region Management" section. This section (which will be moved to the appendix in the next version, to free some space for more advanced analysis) directly tackles the excessive shrinking problem in certain environments.
>
> •	Regarding Question 1: In our experiments, we included a variant of CMA, which is a state-of-the-art (SOTA) genetic algorithm. Additionally, Bayesian methods face challenges similar to the ones discussed regarding Weakness 1.
>
> •	Regarding Question 2: Thank you for the note. We will refine the paper's flow, polish the structure, and upload a revised version.

---

> ### Author Response · Authors · 2024-11-28
> **Response for Reviewer YVhg**
>
> We sincerely thank you once again for your informative review. Your feedback has been invaluable in improving our paper, helping us address misunderstandings and refine our approach to better address the challenges in our method.
>
> In the revised version, we have included an ablation test showcasing the results of the detached Hessian method in Figure 4(b) (page 7). As highlighted, this method significantly reduces computational demands while achieving results nearly identical to those of the original approach. Additionally, we have restructured the paper to improve its flow and readability, ensuring that the presentation is both clearer and more organized.
>
> We hope that these improvements address the key concerns raised and hope the revised version provides a more compelling and polished representation of our work. We deeply value your time and expertise in revisiting the updated manuscript.

---

### Author Response · Authors · 2024-11-18
**Response for reviewers**

**To all reviewers**:

We would like to thank the reviewers for their insightful feedback. To begin, we would like to provide preliminary responses to address key concerns and questions (as separate comments to each review). Over the coming days, we will incorporate your valuable suggestions into a revised version of the paper.

---

### Author Response · Authors · 2024-11-28
**Revised version of the paper**

We want to thank the reviewers for their informative responses, we present a revised version of the paper with several key adjustments:
- Analytical Analysis: We now include a comprehensive analytical analysis of the algorithm, covering gradient accuracy and convergence rate. Additionally, we clarify the implications of this analysis and highlight the differences compared to the analysis presented in the EGL paper. This addition should alleviate concerns regarding the theoretical implications of each version of our algorithm. Additionally, it should also provide guidance for future research.
- Ablation Study: We conducted an ablation study to demonstrate the contribution of the core components of our work: the Hessian corrections and optimistic optimization compared to the technical improvements we applied to EGL. This study clearly shows that the majority of the improvement in the overall score is attributed to our core contribution.

---

### Author Response · Authors · 2024-11-28
**Revised version of the paper**

In the revised version of the paper, we also address these concerns:
- HGL explanation: We provide a more detailed explanation of the rationale behind HGL in the appropriate section. Specifically, we clarify and explain why the Hessian is utilized exclusively during the training phase and is not involved in the actual optimization step.
- Efficient Hessian Computation: We showed that detaching the Hessian from the computation graph and optimizing the network without backpropagating through the Hessian yields similar results. This approach leverages the second-order Taylor expansion more efficiently while significantly reducing memory usage. This important addition addresses the issue of the computational resource required for running hessian correction.

---

### Author Response · Authors · 2024-11-28
**Revised version of the paper**

We wanted to mention a couple of extra points you can look for in the revised version:
- Adversarial Attack: Using the detached hessian mentioned in the previous comment, we conducted an adversarial attack and presented the corresponding results. Thus, presenting the utility of this method in real-world applications.
- We’ve added a more comprehensive study of the success rate of each algorithm. Using a performance graph, we show the success of each algorithm for different accuracy requirements.
- HP Tolerance: We delve deeper into picking the correct hyperparameters. We used the analytical analysis to show guidelines for picking the correct values. This change will allow readers who want to solve their unique problems a method to pick the appropriate hyper-parameters.

---

### Meta-Review · Area_Chair_rgm3 · 2024-12-26

**Metareview:**

The paper propose OHGL (Optimistic Higher order Gradient learning ) that uses Hessian corrections for solving black box optimization problems.This is validated on synthetic benchmark COCO and in adversarial training and code generation.

One of the main concerns with the paper is the computational complexity due to hessian computation and backpropagation through it,
 reviewer 86Mz provided a solution to this problem via detaching hessians from the computational graph, authors explored these ideas on the synthetic benchmark, with a reduction in computational time without incurring errors on the synthetic example.

The paper clarity and rigor improved a lot in the revision and the rebuttal phase, nevertheless integrating speeding up the second order information is crucial for higher dimensional settings and further experimentation on the high dimensional examples  is needed to improve the scalability of the method. We encourage the authors to reproduce those experiments in the high dimensional case and include them in the paper.

**Additional Comments On Reviewer Discussion:**

The paper was largely discussed between authors and reviewers, with  reviewer 86Mz proposing a solution to scale up the method.

---

### Decision · Program_Chairs · 2025-01-22

Reject